# Stability of Hydroxo/Oxo/Fluoro Zirconates vs. Hafniates—A DFT Study

Jennifer Anders [1,*] , Fabian Göritz [1] , Anselm Loges [2] , Timm John [2] and Beate Paulus [1]

1   Institute for Chemistry and Biochemistry, Freie Universität Berlin, Arnimallee 22, 14195 Berlin, Germany
2   Institute for Geological Sciences, Freie Universität Berlin, Malteserstr. 74–100, 12249 Berlin, Germany
*   Correspondence: jennifer.anders@fu-berlin.de

**Abstract:** We performed density functional theory (DFT) calculations on binary and ternary oxo/fluoro crystals of the geochemical twin pair zirconium and hafnium to evaluate and compare their stabilities. This is the first DFT study on bulk $ZrF_4$ or $HfF_4$, as well as on a hypothetical $ZrOF_2$ or $HfOF_2$ bulk crystal. For $\alpha$-$MO_2$, $\beta$-$MF_4$ and $MOF_2$, we have found significantly higher cohesive energies for the respective hafnium species. This suggests a considerable gap in affinity toward fluorine and oxygen between the twin pair in the solid state. In agreement with experimental findings, this gap is slightly more pronounced for fluorine. This study is also the first to evaluate the theoretical, endothermic mono-hydroxylation of the respective fluorides or oxyfluorides to model the difference in affinity toward fluoride versus hydroxide. For these, we could also find a slight energetic preference for the hafnium compound.

**Keywords:** geochemical twins; HFSE; DFT; baddeleyite; fluorozirconate; fluorohafniate; oxofluoride





## 1. Introduction

### 1.1. Motivation

In this paper, we investigate the subtle differences between zirconium and hafnium in the solid state. Both elements form a so-called geochemical twin pair because they behave almost identically throughout most geochemical processes. This twin behavior is a simple result of the equivalent charge to radius ratio of the respective ions in their single stable oxidation state of +IV. Measured in eightfold coordination, as they occur in pure fluorides, the ionic radii are nearly identical with 0.84 Å for Zr to 0.83 Å for Hf [1]. According to these small ionic radii, combined with the high charge, both elements belong to the economically interesting class of high field strength elements (HFSE). Since the past decade, zirconia ($ZrO_2$) and the analogous hafnia ($HfO_2$) attract a lot of attention due to their ferroelectricity, which is well reviewed by Park et al. [2]. As fluorides, both elements are widely applied in optics based on ZBLAN (Zr, Ba, La, Al, Na) fluoride glasses. Depending on the glass composition, the optical window can reach from deep IR to near UV. As ultra thin fibers, ZBLAN fluoride glasses are a well-suited successor of silica in photonics, promising a much larger transmission bandwidth [3,4]. While the main component of these glasses is typically $ZrF_4$, some specialized ones use the analogous $HfF_4$. Doped with cerium, the latter shows excellent scintillating properties [5–7]. A small impurity of the respective other twin element can usually be found in all these Zr/Hf-based materials. The nuclear industry demands extremely pure, and thus very well separated, Zr/Hf because of their opposite thermal neutron-adsorption cross sections. Zr is used for materials with minimal neutron interactions, such as, e.g., for cladding of the nuclear fuel rods. The high absorption of Hf, on the other hand, makes it an ideal material for nuclear control rods [8].

Due to their twin character, fractionating Hf from Zr is not a trivial task. The behavior of ions is typically controlled primarily by their charge and radius (CHARAC) in natural geochemical systems. Therefore, and as a result of their identical charge and nearly identical

radius, Zr and Hf ions usually do not fractionate from each other in nature and are typically found in ratios close to their chondritic ratio of 37:1 in most rocks and minerals [9–11]. The rare exceptions to this rule are fluoride-rich hydrothermal veins and fluoride-rich pegmatitic melts, where much lower Zr:Hf ratios of about 2:1 have been observed [9,12–14]. It has been suggested that the cause for this Hf-enrichment is a slightly higher affinity of Hf to halogens compared to Zr [9,11,15]. The hafnium to halogen bond is generally a bit stronger than the respective zirconium to halogen bond (e.g., 240 meV for the diatomic Zr/Hf fluorides) [16,17]. Computationally, it has has also been shown that the chemical adsorption of gaseous hydrogen fluoride (HF) onto the $HfO_2$ (111) surface is 150–210 meV stronger than for $ZrO_2$ [18]. On an industrial scale, the difference in halogen affinity is exploited to separate Zr/Hf, as the process relies on the isolation of the respective oxochlorides [8,19]. Another pathway to obtain the highest purities uses the fluorides ($MF_4$) formed from the oxides in anhydrous HF (aHF) gas according to reaction (1) [20]. Note that this reaction involves oxofluoride intermediates in varying stoichiometry. These will be further discussed in the following Section 1.2.

$$MO_2 \xrightarrow{\text{aHF}} M_pO_qF_r \xrightarrow{\text{aHF}} MF_4 \qquad \text{with } M = \{Zr, Hf\} \qquad (1)$$

Furthermore, Hf also shows a higher affinity for fluorine in the liquid phase as suggested by solubility experiments in aqueous HF (aq. HF) of low concentrations (1–200 mM) [15,21]. Already at these low F-activities, only the di-fluoro $HfF_2(OH)_2$ complex is found for Hf, while Zr forms mono-fluoro $ZrF(OH)_3$, as well as di-fluoro $ZrF_2(OH)_2$ compounds. At a concentration of 1 molal aq. HF and elevated temperatures of 350 °C, no difference between Zr and Hf in principal complex stoichiometry was observed using X-ray spectroscopic methods, but instead slightly shorter average metal-ligand distances for the Hf complexes compared to those of Zr [22]. This may also suggest a stronger bond of Hf with fluoride. While the complex stoichiometry could not be directly observed, previous studies have shown that electrically neutral difluoro-dihydroxy complexes of Zr and Hf can be expected under these conditions [21,23].

We aim to contribute further insight on what sets the interaction of Zr/Hf to fluorine apart from Zr/Hf to oxygen and, therefore, provide a quantum-chemically-based hypothesis on the observed different solubility by solid $Zr/HfO_2$ in aq. HF. This requires, in a first step, to understand the respective solids. Consequently, this project will first contrast the known $Zr/HfO_2$ and $Zr/HfF_4$ to compare to the hypothetical, mixed $Zr/HfOF_2$. In a second step, we broaden our comparison to hydroxylation products of the respective mixed and pure fluorides. In nature, neither crystalline Zr or Hf hydroxides are known, nor simple compounds that incorporate any OH-group within the lattice. Therefore, we chose the mono-hydroxylated unit cells as a crystalline model with low OH concentration.

### 1.2. Known Crystal Structures

In nature, Zr is usually found as zircon ($ZrSiO_4$) and to a lesser extent as the binary oxide in baddeleyite (monoclinic $\alpha$-$ZrO_2$) or zirconia (tetragonal $\beta$-$ZrO_2$) structure. Its less abundant twin element Hf is found as an impurity within these minerals. At ambient conditions, the oxides crystallize in the baddeleyite-structure ($P2_1/c$, SG 14), which is the stable phase up to ($ZrO_2$) or 2100 °C or 11 GPa ($HfO_2$), respectively [24–27]. There exist many density functional theory (DFT) studies on the phase transitions and other properties of bulk $ZrO_2$ and $HfO_2$ at the LDA [28–35] and/or GGA [18,30,31,36–43] level (see Table S1 for a full overview). However, on the corresponding binary fluorides, only a single computational study was found, which performed the ab initio perturbed ion method and configuration interaction with single excitations on the crystal clusters [44]. This leaves this study as the first DFT evaluation on fluorozirconate or fluorohafniate. $\beta$-$ZrF_4$ or $\beta$-$HfF_4$ are also monoclinic in their low temperature phase ($C2/c$, SG 15), which are stable up to 910 °C [45–48]. When increasing complexity by forming tertiary compounds as M(IV)-oxofluorides, no computational studies could be found and also the availability of measured crystal structures decreases significantly for Zr, while none have

been published for Hf. Two experimental crystal structures with different compositions can be found for $Zr_pO_qF_r$: a cubic ReO$_3$-structured $ZrO_{0.67}F_{2.66}$ ($Pm\bar{3}m$, SG 221) [49,50] and an orthorhombic $Zr_7O_{8.79}F_{10.21}$ ($Pbam$, SG 55) [51]. It should be noted that neither of them could resolve the O/F positions unambiguously. Furthermore, the off-integer anion content suggests a considerable number of defects within the crystal structure. Recently, the Hf-analog with the approximate stoichiometry of $Hf_7O_9F_{10}$ has been found [52]. Again, they found disordered anions. Unfortunately, they could not refine the unit cell parameters distinctly, leaving the crystal structure unresolved. Despite their widely unknown crystal structures, a series of oxofluorides is described as intermediates between the two binary compounds (see reaction (1)). By XRD, the presence of zirconium oxofluorides in three further stoichiometries could be detected: $Zr_3O_2F_8$, $ZrO_{0.33}F_{3.33}$, $ZrO_{1.3}F_{1.4}$, but only one hafnium oxofluoride as $Hf_2OF_6$ [20]. Earlier, $Hf_3O_2F_8$ has also been reported [53]. Unfortunately, no crystal structures are published for any of these intermediates. Not only the measured, but also the calculated crystal structures, are rare for $M_pO_qF_r$. Even the comprehensive database of the Materials Project only provides three Zr and two Hf species [54]. However, none of them follow a stoichiometry to yield the desired, simple electronic structure of M(IV), O($-$II) and F($-$I). Most of them are instead best described as $O_2$ or $O_3$ molecules enclosed between 2D-sheets of crystalline metal fluorides. Expanding the scope further to the hydroxides such as, e.g., Zr/Hf(OH)$_4$, the same problem of unstable crystal structures is encountered. Despite F and OH being isoelectronic, their affinity to water is very different. Thus, in contrast to the crystalline fluorides, the hydroxides form gels. Only if carefully prepared from crystalline, tetragonal [ZrOCl$_2$] ($P\bar{4}2_1c$, SG 114), its basic structural unit of double oxygen-bridged zirconium squares may be retained to some extend. In the formed [Zr$_4$(OH)$_8$(H$_2$O)$_{16}$]Cl$_8$, the neighboring tetramer units are double bridged by a pair of hydroxides [55–57]. However, according to the largest inorganic crystal structure database (ICSD) [58], actual crystalline structures built from the elements Zr, O, F, and H are exclusively made from ZrF$_i$ lattices with crystal water as [ZrF$_4$·(H$_2$O)$_j$] with $j = \{1, 3\}$, or if $i = \{5, 6\}$ with incorporation of oxonium ions [59–64]. No simple crystal structures are known that incorporate hydroxides into the lattice. For this study, we therefore build hypothetical crystalline oxofluorides Zr/HfOF$_2$, as well as hypothetical, mono-hydroxylated species from these structures, as well as from the stable Zr/HfF$_4$. In the search for a suitable model crystal structure with resolved O/F positions and integer stoichiometry for the hypothetical crystalline oxofluoride $M_p^{IV}O_qF_r$, we chose monoclinic TeOF$_2$ ($P2_1$, SG 4) [65]. This choice is based on its stoichiometry and positions of the anions, which precisely describe local metal(IV) environments. In addition, its rather small unit cell is composed of four formula units with a corresponding unit cell volume per formula unit and lays exactly in between the narrow 1.6 Å$^3$ gap of ZrF$_4$ and HfF$_4$. We therefore consider the crystal structure of TeOF$_2$ as the most-fitting approximation to the hypothetical Zr/HfOF$_2$, despite the remaining lone-pair of Te(IV), which also contributes a 0.25 Å bigger ionic radius compared to Zr/Hf when measured at sixfold coordination as present in TeOF$_2$ [1].

## 2. Computational Details

All periodic density functional theory (DFT) calculations have been performed with the Vienna Ab Initio Simulation Package (VASP, software version 5.4.4) [66] running on the supercomputer cluster HLRN in Göttingen, Germany. As the exchange-correlation functional, the generalized gradient approximation (GGA) by Perdew–Burke–Ernzerhof (PBE) has been applied [67]. A published, elaborate benchmark with different functionals (PBE, PBEsol, RPBE, and TPSS) with or without D2 or D3 dispersion correction [68,69] and/or Hubbard-type correction onto Zr-4d revealed that plain PBE performs well on the geometrical data of monoclinic ZrO$_2$. It also showed that applying a Hubbard-type correction onto the conduction band (CB) forming Zr-4d does not improve the results [40]. A detailed discussion on the choice of functional is given in the SI [70–73]. Another paper on ZrO$_2$ and HfO$_2$ found that the effect of dispersion is also negligible for the

surface adsorption of two HF molecules when comparing PBE with or without dispersion correction according to the Tkatchenko–Scheffler scheme [18,74]. Therefore, we did not apply a dispersion correction for our bulk calculations of the ionic solids. Core electrons have been treated by the projector-augmented wave (PAW) method [75,76] using the VASP-inherent potential files O_h, F_h, Zr_sv, Te, and Hf_sv leaving 6, 7, 12, 6, and 12 valence electrons, respectively. The valence electrons have been described by plane waves to a kinetic energy cut-off of 773 eV. For electron smearing, Gaussian smearing with a width of 0.2 eV has been used for all ionic solids and molecular calculations. For the pure metals of Zr and Hf, a convergence test with second-order Methfessel–Paxton smearing with widths of 0.05–0.15 eV in 0.05 eV steps has been completed. A width of 0.05 eV minimized the difference between total energy and free energy for both metals. For solids of $ZrO_2$, $HfO_2$, $ZrF_4$, and $HfF_4$, the $k$-grid convergence has been tested for Monkhorst–Pack $k$-grids of $x \times x \times x$ with $x = \{1, 2, 3, 4, 5, 7, 9\}$. Convergence within 1.5 meV per unit cell in respect to the finest grid was found at $x = 4$ for all compounds. This $k$-grid was then applied on all ionic solids. The pure metals of Zr and Hf have been tested for $k$-grids with $x = \{1, 2, 3, 4, 5, 7, 9, 15, 17\}$ yielding convergence within 2.5 meV per unit cell at $x = 15$ with respect to the finest grid. The molecular calculations of $H_2$, HF, $H_2O$, $O_2$, and $F_2$ have been set up in a cubic box of 25 Å side length at the $\Gamma$-point only to avoid artificial intermolecular interactions. The structural relaxations of all solids have been conducted with the conjugate-gradient algorithm in three subsequent steps with increasing degrees of freedom. At first, only the atomic positions were relaxed. This was followed by the additional relaxation of the unit cell vectors, while keeping the volume fixed. In the third step, the positions, unit cell vectors, and unit cell volume were allowed to adapt simultaneously. The initial start structures for $MF_4$ [45,46], as well as $TeOF_2$ [65], were taken from the experimental crystal structures discussed above in Section 1.2, while the $MO_2$ were initiated from the PW91-relaxed structure [41]. The pure metals started from their respective experimental, hexagonal crystal structure ($P6_3/mmc$, SG 194) [77,78]. To aid convergence, spin polarization was allowed and/or the ionic step width (POTIM) reduced from its default of 0.5 to 0.1 Å if needed. The accurate precision setting was used for all calculations. Geometry relaxations were performed with a self-consistent field (SCF) convergence criteria of 0.01 meV per unit cell and 0.1 meV per unit cell for the difference in total energy between two ionic steps. Final total energies and Bader charges were generated with an SCF criteria of 0.001 meV per unit cell. The atom-in-molecule-derived Bader charges were obtained by the algorithm of the Henkelman group [79–83]. Structures were built with the Python package pymatgen [84] and visualized in VESTA [85].

### 3. Results and Discussion

*3.1. Oxo/Oxofluoro/Fluoro Crystals of Zr vs. Hf*

3.1.1. Crystal Structures

We calculated the crystal structures of the geochemical twin elements Zr/Hf as binary oxides and fluorides, as well as hypothetical 1:2 oxofluorides. Table 1 shows their relaxed conventional unit cell parameters with formula units, lattice vector lengths, off-diagonal angle, and corresponding volumes. For the binary fluorides, there also exists a smaller, primitive unit cell of only six formula units. Thus, all calculations were conducted on the symmetry-translated primitive cells.

Table 1 shows good agreement between the relaxed unit cell parameters of the binary oxides and fluorides and the experimental literature data. The deviation of the unit cell vector length was less than 0.16 Å or 1.3%. The absolute and relative deviations from the experiment for all binary compounds are listed in the SI (see Table S4). For $ZrO_2$ and $HfO_2$, our PBE (cutoff energy of 773 eV) relaxed structures represent the crystal structure even better than the previously reported PB91 results (cutoff energy of 495 eV), which were used as input structures [41]. In accordance with their geochemical twin character, $ZrO_2$ and $HfO_2$ share a practically equivalent structure, just as the respective fluorides $ZrF_4$ and $HfF_4$. The relaxed structures are shown in Figure 1. Within the oxides, all metal centers

are symmetry equivalent and coordinated by seven oxygen atoms, for which two non-symmetry equivalent positions exist. The fluorides contain two metal types, each eightfold coordinated. $MF_4$ contains seven non-symmetry equivalent fluorine positions. Although $MO_2$ and $MF_4$ show a high similarity between the respective Zr and Hf-species, the Hf bond lengths are a bit smaller than the respective Zr bonds. The relaxed Hf–O bond lengths are 2.04–2.23 Å vs. 2.05–2.27 Å for Zr–O. Note that the latter agree perfectly with the measured bond lengths [86]. Within the fluorides, the bond lengths are generally smaller, which goes along with a smaller difference between Zr/Hf. While 2.03–2.13 Å are found for the relaxed Hf–F bonds, the range of bond lengths is increased by only 0.02 Å for Zr–F with 2.05–2.15 Å. It should be noted that the difference of Zr–F to the experimental bond lengths of 2.03–2.18 Å is of the same order of magnitude [45] (see Tables S5 and S6 for all respective bond lengths).

**Table 1.** Relaxed unit cell parameters vs. experimental (lit. exp.) and calculated literature (lit. PB91) values. Given are the number of formula units per unit cell ($N_{\text{f.u.}}$), unit cell vector lengths ($a, b, c$), unit cell volume ($V$), unit cell volume per formula unit ($V_{\text{f.u.}}$), and the non-orthogonal angle ($\beta$); note that $\alpha = \gamma = 90°$.

| Compound | $N_{\text{f.u.}}$ | $a$ (Å) | $b$ (Å) | $c$ (Å) | $V$ (Å$^3$) | $V_{\text{f.u.}}$ (Å$^3$) | $\beta$ (°) |
|---|---|---|---|---|---|---|---|
| $ZrO_2$ | 4 | 5.154 | 5.224 | 5.332 | 141.56 | 35.39 | 99.55 |
| lit. PB91 [41] | 4 | 5.197 | 5.279 | 5.349 | 144.74 | 36.18 | 99.53 |
| lit. exp. [86] | 4 | 5.150 | 5.212 | 5.317 | 140.88 | 35.22 | 99.23 |
| $HfO_2$ | 4 | 5.105 | 5.182 | 5.277 | 137.64 | 34.41 | 99.54 |
| lit. PB91 [41] | 4 | 5.128 | 5.191 | 5.297 | 139.25 | 34.81 | 99.71 |
| lit. exp. [87] | 4 | 5.114 | 5.168 | 5.290 | 138.03 | 34.51 | 99.21 |
| $TeOF_2$ | 4 | 5.212 | 8.025 | 5.485 | 227.81 | 56.95 | 96.82 |
| lit. exp. [65] | 4 | 5.307 | 8.289 | 5.513 | 241.09 | 60.27 | 96.22 |
| $ZrOF_2$ | 4 | 5.305 | 6.879 | 5.318 | 193.88 | 48.47 | 92.63 |
| $HfOF_2$ | 4 | 5.265 | 6.982 | 5.388 | 197.74 | 49.44 | 93.22 |
| $ZrF_4$ | 12 | 11.694 | 9.889 | 7.660 | 710.40 | 59.20 | 126.68 |
| lit. exp. [45] | 12 | 11.845 | 9.930 | 7.730 | 732.53 | 61.04 | 126.32 |
| $HfF_4$ | 12 | 11.609 | 9.816 | 7.600 | 694.85 | 57.90 | 126.65 |
| lit. exp. [46] | 12 | 11.725 | 9.869 | 7.636 | 713.48 | 59.46 | 126.15 |

To investigate a simple hypothetical mixed oxofluoride, the crystal structure for $TeOF_2$ was taken and the metal centers replaced with Zr and Hf, respectively [65]. The Te-structure was selected because of its relatively simple, well-suited crystal structure (see discussion in Section 1.2) and the +IV oxidation state of its metal centers. However, as a p-block element, Te(IV) is left with a 5s-lone pair that leads to a six-fold coordination in the 1:2 mixed oxofluoride, while Zr(IV)/Hf(IV) have a formally vacant valence shell and can assume a higher coordination. Hence, six-fold coordinated Te(IV) has an ionic radius of 0.97 Å, while a significantly smaller radius can be found for Zr(IV) or Hf(IV) with 0.72 Å or 0.71 Å at six-fold and 0.78 Å or 0.76 Å at seven-fold coordination, respectively [1]. The missing lone pair and the overall smaller ionic radii explain that the unit cell of Zr/$HfOF_2$ is significantly smaller in the $b$-direction, more densely packed than $TeOF_2$ and with a $\beta$-angle closer to 90°. Yet, the resulting structure retains the same low-symmetry space group ($P2_1$, SG 4). The relaxed $MOF_2$ unit cells are visualized in Figure 2.

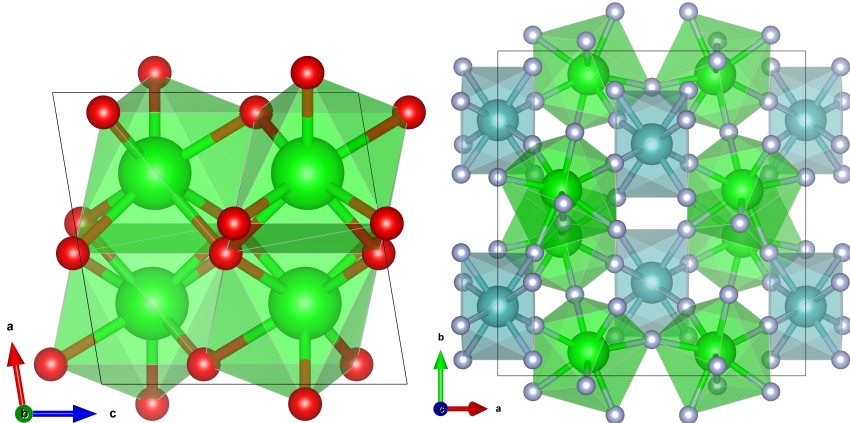

**Figure 1.** Visualized relaxed structures of $ZrO_2$ (**left**) and $ZrF_4$ (**right**). Structures of $HfO_2$ and $HfF_4$ closely resemble the depicted ones in geometry, but with slightly shorter bonds (see Tables S5 and S6). Atoms are colored according to: Zr in green/teal, O in red, F in gray. The fluoride structure has two different metal atoms as basis, shown in teal (center A) and green (center B), further explained in Figure 3 left. Unit cell parameters are given in Table 1.

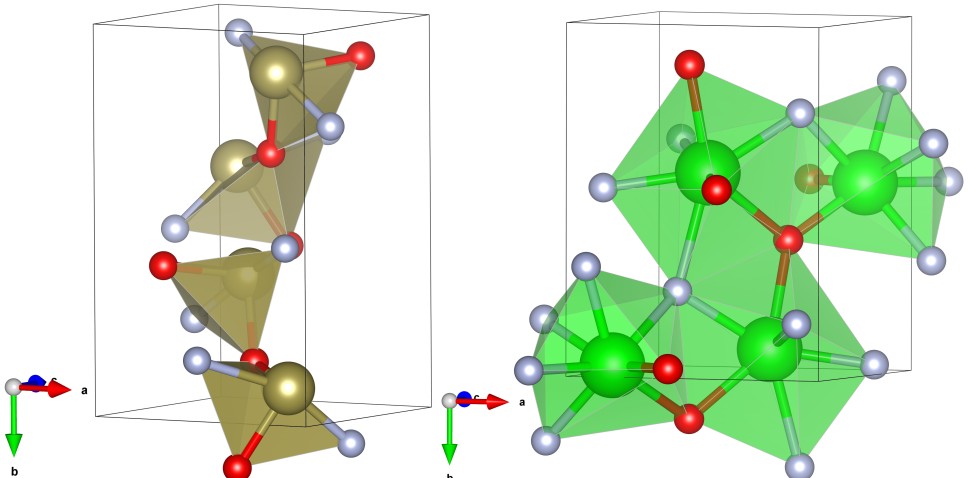

**Figure 2.** Relaxed unit cell of $TeOF_2$ with Te in brown (**left**) and $ZrOF_2$ (**right**). The $HfOF_2$ structure looks equivalent to the Zr-species. Unit cell parameters are given in Table 1.

Each $Zr/HfOF_2$ coordination polyhedron has a seven-fold coordinated metal center of two different types. One center is coordinated to five fluorine and two oxygen atoms, while the other center is coordinated to four fluorine and three oxygen atoms. As seen in Figure 2 (right), one fluorine and one oxygen atom form triple-coordinated bridges that connect the four polyhedra with each other. The unit cell parameters of the input structure $TeOF_2$ and $Zr/HfOF_2$ are shown in Table 1. When comparing the $V_{f.u.}$ listed in Table 1, one finds the binary fluorides to be larger than the binary oxides, in accordance with the two additional atoms per formula unit. The ternary mixed oxofluorides also contain one atom less per formula unit than the fluorides. However, when comparing the experimental $V_{f.u.}$, one finds the original $TeOF_2$ neatly positioned between $HfF_4$ and $ZrF_4$. On the other hand, this does not hold for the relaxed $TeOF_2$, which shrinks by 3 Å$^3$. While the $V_{f.u.}$ of the relaxed $HfF_4$ and $ZrF_4$ is reduced by less than 2 Å$^3$, both hypothetical $HfOF_2$ and $ZrOF_2$ structures shrink by as much as 8 Å$^3$ compared to the relaxed parent Te-structure for reasons discussed above. We found that their relaxed $V_{f.u.}$ agree very well with the respective mean of $MO_2$ and $MF_4$. However, for the oxides and fluorides, we find $\Delta V_{f.u.}(Hf - Zr) = -1$ Å$^3$. We currently have no explanation why the volume of $MOF_2$ does not follow this otherwise observed trend of smaller Hf than Zr compounds. Instead, $HfOF_2$ is 1 Å$^3$ bigger in $V_{f.u.}$ than its Zr-counterpart.

### 3.1.2. Reaction Energies

To analyze the stability of the metal oxides/fluorides and oxofluorides, the cohesive energies ($\Delta E_{\text{coh.}}$) listed in Table 2 have been calculated using the electronic energies of the relaxed structures at 0 K.

**Table 2.** Calculated PBE bulk solid-state cohesive energies ($\Delta E_{\text{coh.}}$) per formula unit (f.u.) for formations of different Zr/Hf-species of oxides (I), fluorides (II), and oxofluorides (III). Below, the literature values [18] for the respective bulk solid-state to gas-phase reactions (s→g) are given.

| | Solid-State Reactions | | | $\Delta E_{\text{coh.}}$ (eV/f.u.) | |
|---|---|---|---|---|---|
| **Nr.** | | | | **M = Zr** | **M = Hf** |
| (I$_a$) | $M_{(s)} + O_{2(g)}$ | $\longrightarrow$ | $MO_{2(s)}$ | −19.279 | −19.545 |
| (I$_b$) | $M_{(s)} + 2\,H_2O_{(g)}$ | $\longrightarrow$ | $MO_{2(s)} + 2\,H_{2(g)}$ | −5.274 | −5.540 |
| (II$_a$) | $M_{(s)} + 2\,F_{2(g)}$ | $\longrightarrow$ | $MF_{4(s)}$ | −18.439 | −18.718 |
| (II$_b$) | $M_{(s)} + 4\,HF_{(g)}$ | $\longrightarrow$ | $MF_{4(s)} + 2\,H_{2(g)}$ | −7.448 | −7.727 |
| (II$_c$) | $MO_{2(s)} + 4\,HF_{(g)}$ | $\longrightarrow$ | $MF_{4(s)} + 2\,H_2O_{(g)}$ | −2.174 | −2.188 |
| (II$_d$) | $MOF_{2(s)} + 2\,HF_{(g)}$ | $\longrightarrow$ | $MF_{4(s)} + H_2O_{(g)}$ | −1.283 | −1.302 |
| (III$_a$) | $M_{(s)} + H_2O_{(g)} + 2\,HF_{(g)}$ | $\longrightarrow$ | $MOF_{2(s)} + 2\,H_{2(g)}$ | −6.165 | −6.425 |
| (III$_b$) | $MO_{2(s)} + F_{2(g)}$ | $\longrightarrow$ | $MOF_{2(s)} + \frac{1}{2}\,O_{2(g)}$ | +0.616 | +0.622 |
| (III$_c$) | $MO_{2(s)} + 2\,HF_{(g)}$ | $\longrightarrow$ | $MOF_{2(s)} + H_2O_{(g)}$ | −0.891 | −0.885 |
| | lit. PBE solid to gas reactions [18] | | | | |
| (II$_c^{s \to g}$) | $MO_{2(s)} + 4\,HF_{(g)}$ | $\longrightarrow$ | $MF_{4(g)} + 2\,H_2O_{(g)}$ | −1.14 | −0.91 |
| (III$_c^{s \to g}$) | $MO_{2(s)} + 2\,HF_{(g)}$ | $\longrightarrow$ | $MOF_{2(g)} + H_2O_{(g)}$ | +2.96 | +3.87 |

A previous computational surface study has shown that the Zr to Hf difference for bulk reactants of $ZrO_2$ and $HfO_2$ is practically equivalent when comparing the purely electronic $\Delta E_{\text{coh.}}$ or when comparing the zero point energy (ZPE) corrected free energies at 500 K ($\Delta G^{500\,\text{K}}$) including the volume work, the temperature-dependent enthalpic terms, as well as translational entropy [18]. The same study also found that for the corresponding (111) surfaces, a small difference between $\Delta E_{\text{coh.}}(Zr-Hf)$ and $\Delta G^{500\,\text{K}}(Zr-Hf)$ occurs due to the additional surface entropy. However, even for the surfaces, this difference is one order of magnitude smaller than $\Delta E_{\text{coh.}}(Zr-Hf)$ itself. Published phonon spectra of $ZrO_2$ and $HfO_2$ show that their ZPE differ by as little as 0–3 meV per formula unit [33]. Measurements and calculations have shown that the shorter bond distances and thus higher force constants of Hf−O compared to Zr−O counterbalance the mass increase of the cation [33,88]. No phonon calculations exists for $ZrF_4$ and $HfF_4$. However, we expect the even stronger mass increase per single M−F bond to be counterbalanced by the additional anion-dominated vibrations with shorter interatomic Hf−F than Zr−F distances (see Section S6 in the SI for a more detailed discussion) [89]. Consequently, we consider the purely electronic quantity of $\Delta E_{\text{coh.}}$ sufficient to evaluate the difference between the respective bulk Zr/Hf-compounds. The values from Table 2 show that hafnium tends to have a stronger affinity towards oxygen and fluorine, as all cohesive energies of solid-state reactions I–III are stronger for the Hf-species than for the corresponding Zr compound. The energetic difference of $\Delta E_{\text{coh.}}(Zr-Hf)$ is 266 meV/f.u. for the binary oxides (see I$_{a-b}$). For the binary fluorides, it is 279 meV/f.u. (see II$_{a-b}$), which means that the Zr vs. Hf difference in the fluorides is by a small amount of 13 meV/f.u. stronger than in oxides. That this value is reproduced within 1 meV/f.u. when forming from the elements or the binary oxides, indicates that electronic energies of the pure metals, as well as the ionic binary compounds are described well enough by the applied computational setup. The hypothetical 1:2 mixed oxofluoride shows similar-sized $\Delta E_{\text{coh.}}(Zr-Hf)$ of 260 meV/f.u. when formed from the elements (see III$_a$). It should be noted that this Zr vs. Hf difference is not somewhere between the binary oxide or fluoride. Instead, it is the smallest and 6 meV/f.u. less than for the oxide. This might correlate with the odd volume trend of $MOF_2$ discussed above. When

comparing the solid-state reactions $II_c$ and $III_c$ with the respective published bulk reactant to molecular gas-phase product reactions $II_c^{s \to g}$ and $III_c^{s \to g}$ [18], one finds the opposite Zr vs. Hf trend. In opposition to the aforementioned trend, the formation of molecular $HfF_{4(g)}$ is about 230 meV/f.u less favored than for the respective $ZrF_{4(g)}$ (see $II_c^{s \to g}$). Even more significant is the deviation in $\Delta E_{coh.}(Zr-Hf)$ for the formation of molecular $HfOF_{2(g)}$ (see $III_c^{s \to g}$). Here, the formation of $MOF_{2(g)}$ is by about 910 meV/f.u. more endothermic for the Hf-species. Accordingly, the stability of Zr and Hf compounds has to be reversed for the molecular products, for reasons that remain unclear up to now. It seems at hand that the missing ZPE correction significantly alters the gap between the two elements of different mass. On the contrary, almost the same values are obtained for the ZPE-corrected $\Delta G^{500\,K}(Zr-Hf)$ with ca. 240 meV/f.u. ($MF_{4(g)}$) and 910 meV/f.u. ($MOF_{2(g)}$), respectively. It may be speculated whether the higher coordination of the metal center or the ligands play a further role. To investigate whether this inconsistency between bulk to bulk and bulk to gas phase reactions is caused by the gas-phase products, we performed test gas-phase calculations. Mullins et al. calculated their gas-phase ZPE with Turbomole/PBE/def-TZVPP. We performed ORCA [90]/PBE/def2-TZVP [91] test calculations with the default effective core potentials [92] or with the ZORA Hamiltonian [93,94] on all electrons. We find that only in the latter, the correct behavior of smaller $Hf-F$ than $Zr-F$ bond distances is reproduced (see Table S8 in the SI). As a consequence, we raise the question of whether a scalar relativistic treatment of $HfF_4$ or $HfOF_2$ is necessary to obtain the right gas-phase values. However, as this study focuses on the Zr/Hf differences within the crystal phase, we leave this question open for further studies.

### 3.2. Mono-Hydroxylated Oxofluoro/Fluoro Crystals of Zr vs. Hf

In the previous subsection, we compared the binary oxides and fluorides, as well as the mixed oxofluorides of Zr vs. Hf. Now, we expand the comparison to the hydroxyl group, isoelectronic to fluorine. As the pure hydroxides of Zr and Hf are too hygroscopic to form crystals, analyzing a theoretical, crystalline $M(OH)_4$ is not meaningful. It is, however, plausible to consider the hydroxylation as a defect of the stable binary $MO_2$ or $MF_4$. Within the case of oxides, each oxygen should be replaced by two hydroxyl groups to remain the metal oxidation state of +IV. This might alter the local crystal structure considerably. Moreover, since we are focusing on the differences in Zr vs. Hf affinities toward O vs. F, replacing O by OH is not target-aimed. Therefore, we only consider the substitution of F by OH. In this case, there is also no issue in generating the mono-hydroxylated species. In this theoretical substitution reaction (2), a single fluorine of the unit cell is substituted by a hydroxyl group.

$$M_6F_{24(s)} + H_2O_{(g)} \xrightarrow{\Delta E_{OH}} M_6F_{23}OH_{(s)} + HF_{(g)} \qquad \text{with } M = \{Zr, Hf\} \tag{2}$$

Analogously, we also consider the mono-hydroxylation of the mixed oxofluorides according to reaction (3).

$$M_4O_4F_{8(s)} + H_2O_{(g)} \xrightarrow{\Delta E_{OH}} Hf_4O_4F_7OH_{(s)} + HF_{(g)} \qquad \text{with } M = \{Zr, Hf\} \tag{3}$$

#### 3.2.1. Crystal Structures

The respective relaxed unit cell parameters of the mono-hydroxylated products according to reaction (2) and (3) are summarized in Table 3. Note that only the most stable positional isomer product is given. Differences between these isomers are discussed in the following Section 3.2.2.

**Table 3.** Relaxed unit cell parameters for the mono-hydroxylated products. Given are the number of pseudo-formula units per unit cell when not differentiating between F and OH ("$N_{\text{f.u.}}$"), unit cell vectors ($a, b, c$), unit cell volume ($V$), unit cell volume per pseudo-formula unit ("$V_{\text{f.u.}}$"), and the non-orthogonal angles ($\beta$); note that all structures possess *P*1 symmetry.

| Compound | "$N_{\text{f.u.}}$" | $a$ (Å) | $b$ (Å) | $c$ (Å) | $V$ (Å$^3$) | "$V_{\text{f.u.}}$" (Å$^3$) | $\alpha$ (°) | $\beta$ (°) | $\gamma$ (°) |
|---|---|---|---|---|---|---|---|---|---|
| $Zr_6F_{23}OH$ | 6 | 7.649 | 7.666 | 7.681 | 355.54 | 59.26 | 117.43 | 80.48 | 117.02 |
| $Hf_6F_{23}OH$ | 6 | 7.591 | 7.610 | 7.626 | 347.97 | 58.00 | 117.41 | 80.40 | 117.01 |
| $Zr_4O_4F_7OH$ | 4 | 5.257 | 5.371 | 6.830 | 192.67 | 48.17 | 90.49 | 90.22 | 92.24 |
| $Hf_4O_4F_7OH$ | 4 | 5.289 | 5.372 | 6.830 | 193.84 | 48.46 | 90.23 | 90.37 | 92.71 |

Table 3 also contains the number of pseudo-formula units per unit cell ("$N_{\text{f.u.}}$") and the resulting volume per pseudo-formula unit ("$V_{\text{f.u.}}$"), which is obtained when not differentiating between the fluorine atoms and the hydroxyl group. We use these numbers to compare the volume of the mono-hydroxylated species to the respective crystalline reactants. Compared to the binary fluorides, $V_{\text{f.u.}}$ remains equivalent during mono-hydroxylation for both $M_6F_{23}OH$ (see Table 1). Looking at the oxofluorides, both mono-hydroxylated species shrink a bit. However, while for $Zr_4O_4F_7OH$, this is only very marginally the case with 0.3 Å$^3$, it is 1.0 Å$^3$ for $Hf_4O_4F_7OH$. This may be connected to the unexpected large volume of $HfOF_2$, being bigger than the respective Zr-species by also 1.0 Å$^3$, while the Hf-species are otherwise smaller by 1.0 Å$^3$ ($MO_2$) or 1.3 Å$^3$ ($MF_4$).

### 3.2.2. Reaction Energies and H-Bond Patterns

We use the electronic energy contribution ($\Delta E_{\text{OH}}$) to reactions (2) and (3) as given in Table 4 as a measure to quantify the difference in affinity of the geochemical twins between fluorine and oxygen in the form of a hydroxyl group.

**Table 4.** Calculated electronic energy contribution ($\Delta E_{\text{OH}}$) per unit cell (U.C.) for formations of mono-hydroxylated products according to reactions (2) or (3).

| | Solid-State Reactions | | | $\Delta E_{\text{OH}}$ (eV/U.C.) | |
|---|---|---|---|---|---|
| Nr. | | | | M = Zr | M = Hf |
| (2) | $M_6F_{24(s)} + H_2O_{(g)}$ | $\longrightarrow$ | $M_6F_{23}OH_{(s)} + HF_{(g)}$ | +0.642 | +0.628 |
| (3) | $M_4O_4F_{8(s)} + H_2O_{(g)}$ | $\longrightarrow$ | $M_4O_4F_7OH_{(s)} + HF_{(g)}$ | +0.708 | +0.649 |

The binary fluorides possess two non-symmetry-equivalent metal centers, labeled A and B, which are both coordinated by eight fluorine atoms. Seven non-symmetry-equivalent fluorine atoms exist (see Figure 3 left). Metal A is coordinated by fluorine positions 1–4, each twice. Metal B is coordinated by all seven fluorine positions, with only position 7 being doubly coordinated. Each two metal centers are connected via one bridging fluorine atom. A–B by fluorine 1 and B–B by fluorine 7, while A is not neighbored by another A. Mono-hydroxylation at any of these non-symmetry equivalent positions was considered. For positions 1–4 and 7, two starting M–O–H angles were tested.

The most stable conformation for each OH position is listed in Table 5. For both Zr and Hf, a hydroxylation at position 2 was found to yield the most stable reaction product. Its structure is visualized in Figure 4. However, the formation of the most stable mono-hydroxylated product according to reaction (2) was endothermic by about 0.6 eV for both metals. This is in line with the absence of the known crystal structures of Zr or Hf that actually incorporates OH (see Section 1.2). Nevertheless, already in this mono-hydroxylation of one out of 24 fluorine atoms, a small difference of 14 meV/U.C. in $\Delta E_{\text{OH}}$ was found between $Zr_6F_{23}OH$ (642 meV/U.C.) and $Hf_6F_{23}OH$ (628 meV/U.C.). This subtle energetic difference corresponds to 160 K. Practically the same energetic difference of 13 meV/f.u., favoring Hf over Zr, was found for the fluorination of the binary oxide $MO_2$ to $MF_4$ (see Table 2).

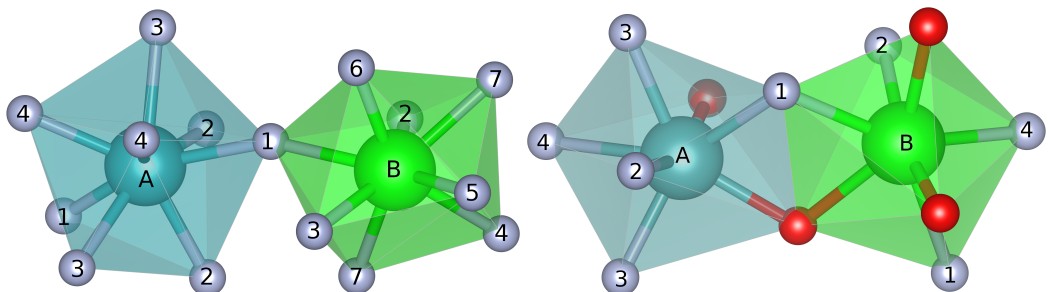

**Figure 3.** Visualization of the two distinct coordination polyhedrons A and B found in $MF_4$ (**left**) and $MOF_2$ (**right**). The non-symmetry equivalent fluorine atoms are labeled accordingly.

X-ray spectroscopic experiments at elevated temperatures in 1 molal aq. HF solutions have shown slightly lower average radial distances between central atom and the ligands for $HfF_2(OH)_2 \cdot 2\,H_2O$, compared to $ZrF_2(OH)_2 \cdot 2\,H_2O$, in a temperature window between 200 and 300 °C [22]. In light of the findings of the present contribution, this difference (e.g., 1.969 vs. 1.990 Å at 300 °C) can be interpreted to be the result of slightly tighter binding of the F and OH ligands to Hf than to Zr, or higher affinity of Hf for F.

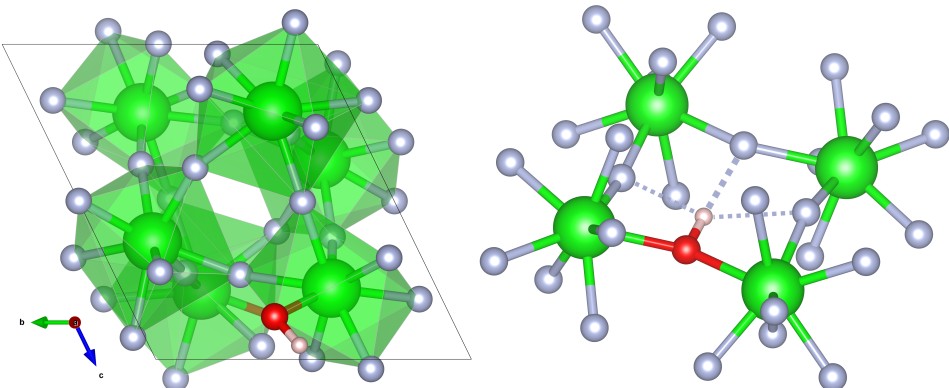

**Figure 4.** Relaxed unit cell of the most stable $Zr_6F_{23}OH$ structure obtained by mono-hydroxylation at position 2 (**left**). The $Hf_6F_{23}OH$ structure looks equivalent to the Zr species. The unit cell parameters are given in Table 3. The structural insert (**right**) shows the typical trifurcated H-bond with a main component (bold dashes) and two minor components (thin dashes). The structural H-bond data is listed in Table 5.

It was found in all calculated $M_6F_{23}OH$ structures that the hydroxyl group is oriented toward a polyhedral gap minimizing steric hindrance. Moreover, substitution of a fluorine atom by a hydroxyl group causes additional interaction between the added hydrogen and the surrounding fluorides by hydrogen bonding. The geometrical H-bond data is summarized in Table 5.

In all structures a trifurcated H-bond was observed. This means that one H atom establishes H-bond interactions to three different [95], yet closely located, F atoms inside a tetrameric Zr/Hf structure. A visualization of this structural motive is presented in Figure 4 (right). The major component faces the opposite side with a rather broad angle of 143–178°. The two minor components are located at the left and right. Their angles are much sharper with 99–119°. However, in most isomers, their distances are shorter than for the main component. We applied the H-bond classification introduced by Jeffrey to distinguish their main and minor components into strong, moderate, and weak H-bonds [96]. No single H-bonds may be regarded as strong and only the main component of some positional isomers classify as moderate H-bonds. All others are considered weak, as they either have a distance longer than 2.2 Å or an angle sharper than 130°. These are marked by parenthesis in Tables 5 and 6. Despite this general, non-solid-state-specific classification, the published mean $O-H\cdots F$ angles within transition metal compounds are 130–160° [97].

This is already much wider than for the respective other halogens. Compared to the mean O$-$H$\cdots$F angles, the main H-bond components in Zr/Hf$_6$F$_{23}$OH already suggest the presence of rather strong O$-$H$\cdots$F interactions for a crystalline compound. The lowest-in-energy structure (with position 2 OH-substituted) also possesses the shortest H-bond length for the main component with 2.09 Å for Zr or 2.10 Å for Hf, respectively. Additionally, it is nearly linear with an angle of 178° for Zr or 174° for Hf. This indicates a correlation with the overall energy of the structure. It should be noted that this is very close to the H-bond angle of 176° found within an infinite HF-chain [98,99]. Even though there are other positional isomers, with likewise wide angles of up to 178° for their major component, their H$\cdots$F distances are much longer. Consequently, their total energies are also higher. It may be noted that $R_{\text{O}-\text{H}}$ does not significantly change within any structure and also does so only marginally within the mono-hydroxylated oxofluorides of M$_4$O$_4$F$_7$OH (see Table 6).

**Table 5.** Position of hydroxylation (OH Pos.), energy difference per unit cell (U.C.) between the positional isomers with respect to the most stable isomer ($\Delta E_{\text{Pos.}}$), hydroxyl group bond length ($R_{\text{O}-\text{H}}$), H-bond length ($R_{\text{H}\cdots\text{F}}$), and angle ($\angle_{\text{O}-\text{H}\cdots\text{F}}$) within the mono-hydroxylated zirconium fluoride (top) or hafnium fluoride (bottom); H-bonds classified as weak by distance or angle are given in parenthesis.

| | **Zr$_6$F$_{23}$OH** | | | |
|---|---|---|---|---|
| **OH Pos.** | **$\Delta E_{\text{Pos.}}$ (eV/U.C.)** | **$R_{\text{O}-\text{H}}$ (Å)** | **$R_{\text{H}\cdots\text{F}}$ (Å)** | **$\angle_{\text{O}-\text{H}\cdots\text{F}}$ (°)** |
| 1 | 0.168 | 0.97 | (2.53), (2.17), (2.05) | (164), (101), (107) |
| 2 | 0 | 0.98 | 2.09, (2.07), (2.11) | 178, (104), (107) |
| 3 | 0.091 | 0.98 | 2.11, (2.16), (2.14) | 154, (106), (103) |
| 4 | 0.137 | 0.98 | (2.46), (1.92), (1.86) | (176), (112), (110) |
| 5 | 0.050 | 0.98 | 2.17, (2.04), (2.18) | 167, (107), (106) |
| 6 | 0.227 | 0.98 | (2.59), (2.02), (1.84) | (169), (110), (116) |
| 7 | 0.188 | 0.97 | (2.46), (2.04), (1.98) | (143), (99), (107) |
| | **Hf$_6$F$_{23}$OH** | | | |
| **OH Pos.** | **$\Delta E_{\text{Pos.}}$ (eV/U.C.)** | **$R_{\text{O}-\text{H}}$ (Å)** | **$R_{\text{H}\cdots\text{F}}$ (Å)** | **$\angle_{\text{O}-\text{H}\cdots\text{F}}$ (°)** |
| 1 | 0.174 | 0.97 | 2.20, (2.01), (2.18) | 170, (109), (99) |
| 2 | 0 | 0.97 | 2.10, (2.12), (2.04) | 174, (106), (104) |
| 3 | 0.091 | 0.97 | 2.11, (2.15), (2.13) | 153, (106), (103) |
| 4 | 0.156 | 0.97 | (2.44), (1.91), (1.82) | (178), (112), (111) |
| 5 | 0.049 | 0.98 | 2.17, (2.04), (2.17) | 164, (107), (106) |
| 6 | 0.273 | 0.98 | (2.38), (2.07), (1.76) | (168), (104), (119) |
| 7 | 0.093 | 0.97 | 2.11, (2.12), (2.15) | 154, (104), (106) |

Just as with the mono-hydroxylated fluorides, all non-symmetry-equivalent positions were considered (see Figure 3 right). The H-bonds have been analyzed likewise. In contrast to these, however, no reoccurring H-bond motive within all positional isomers was found. The positional isomers seem more diverse. What remains is the high similarity between the Zr and Hf-species. In the mono-hydroxylated oxofluorides, H-bonds can form toward oxygen or fluorine. As seen in Table 6, all positional isomers form only one moderate H-bond component. For all but isomer 3, the H-bond formed towards fluorine. The number of additional weak H-bond components varies between one (isomer 3), two (isomer 1, 4), and three (isomer 2). We note that the number of H-bond interactions does not seem to correlate with the energetic order of the positional isomers. Figure 5 (right) shows the H-bond pattern of the most stable isomer. Its trifurcated H-bond is not clearly separable into a single main component with two minor components. Despite only one O$-$H$\cdots$F bond classifying as moderate, the absolute difference to the other O$-$H$\cdots$F bond is not as large as in the mono-hydroxylated fluorides. Moreover, the next-stable isomer 3 is only 24 meV/U.C. less stable, but shows a very different H-bond pattern that only consists of two O$-$H$\cdots$O interactions.

**Table 6.** Position of hydroxylation (OH Pos.), energy difference per unit cell (U.C.) between the positional isomers with respect to the most stable isomer ($\Delta E_{Pos.}$), hydroxyl group bond length ($R_{O-H}$), H-bond length ($R_{H\cdots X}$), and angle ($\angle_{O-H\cdots X}$) within the mono-hydroxylated zirconium oxofluoride (top) or hafnium oxofluoride (bottom); H-bonds classified as weak by distance or angle are given in parenthesis.

| $Zr_4O_4F_7OH$ | | | | | | |
|---|---|---|---|---|---|
| OH Pos. | $\Delta E_{Pos.}$ (eV/U.C.) | $R_{O-H}$ (Å) | $R_{H\cdots F}$ (Å) | $\angle_{O-H\cdots F}$ (°) | $R_{H\cdots O}$ (Å) | $\angle_{O-H\cdots O}$ (°) |
| 1 | 0 | 0.99 | 1.82, (2.21) | 147.88, (130.59) | (2.98) | (137.81) |
| 2 | 0.215 | 0.98 | 1.80, (2.29) | 147.70, (99.21) | (2.20), (2.66) | (111.38), (100.00) |
| 3 | 0.024 | 1.00 | — | — | 1.72, (2.50) | 166.17, (115.74) |
| 4 | 0.251 | 0.97 | 1.90, (2.33) | 155.97, (98.93) | (2.78) | (95.05) |

| $Hf_4O_4F_7OH$ | | | | | | |
|---|---|---|---|---|---|
| OH Pos. | $\Delta E_{Pos.}$ (eV/U.C.) | $R_{O-H}$ (Å) | $R_{H\cdots F}$ (Å) | $\angle_{O-H\cdots F}$ (°) | $R_{H\cdots O}$ (Å) | $\angle_{O-H\cdots O}$ (°) |
| 1 | 0 | 0.99 | 1.87, (2.37) | 147.41, (131.32) | (2.91) | (140.75) |
| 2 | 0.248 | 0.98 | 1.87, (2.32) | 150.33, (96.44) | (2.21), (2.61) | (112.13), (107.42) |
| 3 | 0.022 | 1.00 | — | — | 1.71, (2.56) | 165.59, (116.82) |
| 4 | 0.284 | 0.97 | 1.95, (2.28) | 162.72, (98.86) | (2.46) | (91.40) |

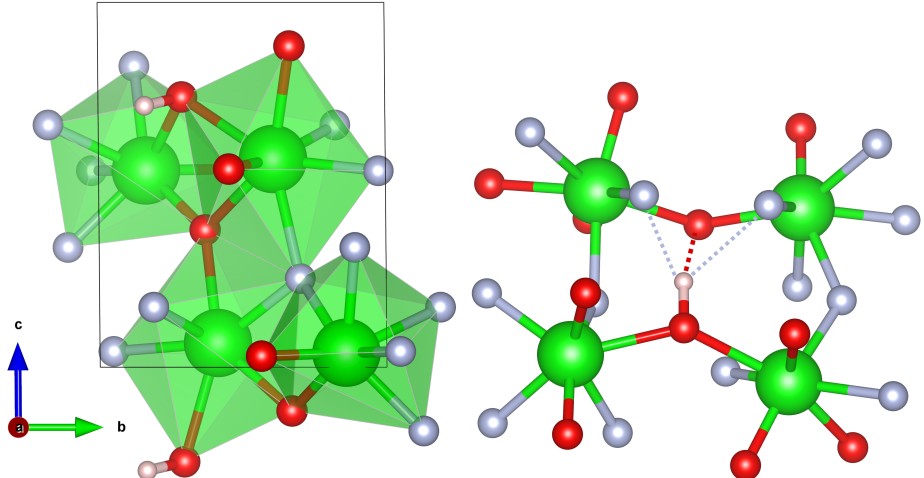

**Figure 5.** Relaxed unit cell of the most stable $Zr_4O_4F_7OH$ structure obtained by the mono-hydroxylation at position 1 (**left**). The $M_4O_4F_7OH$ structure looks practically equivalent to the Zr-species. The unit cell parameters are given in Table 3. The structural insert (**right**) shows the trifurcated H-bond with three components (dashes). The structural H-bond data is listed in Table 5.

## 3.3. Partial Charges

To compare the electronic structures, Bader charges were calculated. Within the $MO_2$, we find $-1.3$ e for oxygen and 2.6 e for Zr, as well as Hf. In $MF_4$, the metal centers were slightly more charged by 2.8 e, with $-0.7$ e for fluorine. This marginally more positive charge of the metal centers in the fluorides indicates a slightly higher ionic character for the metal-F-bond compared to the oxides. The situation is similar in the oxofluorides, but two slightly different charges of $-(1.2–1.3)$ e for oxygen and fluorine $-(0.7–0.8)$ e were found. The metal centers have a Bader charge of 2.7 e, which is equivalent to the mean value of metal charges within the pure oxide and fluoride. At this point, we already compare the mono-hydroxylated species for their partial charges. Within $M_6F_{23}OH$, we find $-1.2$ e for oxygen, $-0.7$ e for fluorine, 0.6 e for hydrogen, and 2.8 e for both Zr and Hf. For $M_4O_4F_7OH$, the Bader charges are varying a bit more, however they all range between $-(1.1–1.3)$ e for oxygen, $-(0.7–0.8)$ e for fluorine, 0.6–0.7 e for hydrogen, and 2.6–2.7 e for both metals. Consequently, we cannot see differences in the Bader charges between the geochemical twin pair for any of the calculated crystals. Moreover, the effect of the

anions onto the metal center is rather subtle with max. 0.2 e. Furthermore, no significant differences are detectable after mono-hydroxylation; even the hydroxyl oxygen charge resembles the other oxygen atoms.

## 4. Conclusions

In this paper, we investigated the subtle differences in the solid state between the geochemical twin elements Zr/Hf as binary oxides and fluorides, as well as the hypothetical 1:2 mixed oxofluorides. We found that the cohesive energies to form any of these three products from the elements is significantly larger for hafnium than for zirconium. Consequently, within the solid-state, hafnium seems to have a considerably higher affinity for oxygen as well as for fluorine. However, as shown by the fluorination of the respective oxides, the affinity gap between the geochemical twins is sightly larger for fluorine. This might explain the different solubility products of the oxides in diluted aq. HF. In the second part of this study, we explored the hypothetical, endothermic mono-hydroxylated fluorides and oxofluorides of both metals. For the mono-hydroxylated fluorides, interesting trifurcated H-bonds toward three fluorines were found. Although zirconium and hafnium fluorides prefer the same position for hydroxylation, they do show a small energetic difference, again slightly favoring the reaction with the hafnium species. On the other side, according to their Bader charges, no significant difference between Zr/Hf could be found within any of these compounds. Their partial charges hardly changed, whether they were coordinated with the oxygen of fluorine. Consequently, the topology of their electron density is hardly affected.

**Supplementary Materials:** The following supporting information can be downloaded at: https://www.mdpi.com/article/10.3390/inorganics10120259/s1, Figure S1: $k$-grid convergence for ionic unit cells. Plotted is the difference in total energy per unit cell to the finest grid ($\Delta E_0$). The convergence area of $\Delta E_0 \leq 1.5$ meV is visualized by two horizontal lines. Values for $n = 1, 2$ are not included in this zoomed-in plot due to their high deviation. Figure S2: $k$-grid convergence for metallic unit cells. Plotted is the difference in total energy per unit cell to the finest grid ($\Delta E_0$). The convergence area of $\Delta E_0 \leq 2.5$ meV is visualized by two horizontal lines. Values for $n = 1, 2$ are not included in this zoomed-in plot due to their high deviation. Figure S3: Second-order Methfessel–Paxton smearing convergence for different smearing widths ($\sigma$) in meV per unit cell. Plotted is the difference of free energy at 0 K ($F$) minus the total energy ($E_0$). Figure S4: Bonding situation in a subunit of Zr/HfO$_2$. The oxygen atoms are labeled in accordance with the bond lengths given in Table S5. Figure S5: Bonding situation in a subunit of Zr/HfF$_4$. The fluorine atoms are labeled in accordance with the bond length in Table S6. Table S1: Literature overview on calculated bulk MO$_2$ and MF$_4$ monoclinic crystals with M = Zr or Hf. For plain wave calculations the cutoff energy is given in eV. The $k$-grid is given as $k_a$x$k_b$x$k_c$ or if not available in number of irreducible $k$-points ($k_{\text{irred.}}$). The main aims are abbreviated as phase transitions (PT), band structure or gap (BS), elastic (EP), and dielectric properties (DP) or phonon spectra (Ph). Table S2: $k$-grid converged total energies ($E_0$) in eV for ionic unit cells. Given are the values for the finest grid, the within 1.5 meV converged grid size, and their difference ($\Delta E_0$). The latter is also plotted in Figure S1. Table S3: $k$-grid converged total energies ($E_0$) in eV for metallic unit cells. Given are the values for the finest grid, the within 2.5 meV converged grid size, and their difference ($\Delta E_0$). The latter is also plotted in Figure S2. Table S4: Relaxed unit cell parameters versus experimental (lit. exp.) and calculated literature (lit. PB91) values. Given are the unit cell vector lengths ($a, b, c$), unit cell volume ($V$), unit cell volume per formula unit ($V_{\text{f.u.}}$), and the non-orthogonal angle ($\beta$), each relaxed parameter is also given as absolute difference to the experimental literature value ($\Delta$ exp.) and the deviation from experiment in percentage ($\Delta_{\%}$ exp.); note that $\alpha = \gamma = 90°$. Table S5: Bond lengths between metal center and oxygen atoms ($R_{\text{M-O}}$) in relaxed ZrO$_2$/HfO$_2$ with the absolute difference in bond length between the two M-species ($\Delta R_{\text{M-O}}$). For comparison, also the experimental bond lengths for ZrO$_2$ $\left( R_{\text{Zr-O}}^{\text{exp.}} \right)$ [86] are given together with the absolute difference to the relaxed values $\left( \Delta R_{\text{Zr-O}}^{\text{exp.}} \right)$. All values are given in Å. Table S6: Bond lengths between metal center and fluorine atoms ($R_{\text{M-F}}$) in relaxed ZrF$_4$/HfF$_4$ with the absolute difference in bond length between the two M-species ($\Delta R_{\text{M-F}}$). For comparison, also the experimental bond lengths for ZrF$_4$ $\left( R_{\text{Zr-F}}^{\text{exp.}} \right)$ [45] are given together with the absolute difference to the relaxed



values $\left(\Delta R_{Zr-F}^{exp.}\right)$. All values are given in Å. Table S7: ZPE energies per formula unit (f.u.) obtained from the published optical phonon frequencies at the Γ-point calculated with the PBE, LDA, or PW91 functional. Table S8: Comparison of M–F bond length in Å and ZPE in meV per formula unit for molecular $ZrF_4$ and $HfF_4$ with or without ZORA. References [25,27–48,68–73,77,78,86–94] are cited in the supplementary materials.

**Author Contributions:** Conceptualization, A.L., T.J., J.A. and B.P.; formal analysis, F.G. and J.A.; funding acquisition, B.P.; investigation, F.G. and J.A.; resources, B.P.; supervision, J.A. and B.P.; validation, J.A.; visualization, F.G.; writing—original draft, J.A. and F.G.; writing—review and editing, J.A., F.G., A.L., T.J. and B.P. All authors have read and agreed to the published version of the manuscript.

**Funding:** The project was funded by the Freie Universität Berlin and the Deutsche Forschungs-gemeinschaft (DFG, German Research Foundation) – Project-ID 387284271 – CRC 1349—fluorine-specific interactions.

**Institutional Review Board Statement:** Not applicable.

**Informed Consent Statement:** Not applicable.

**Data Availability Statement:** See supplementary materials. Further data can be requested from the authors.

**Acknowledgments:** The authors thank the North-German Supercomputing Alliance (Norddeutscher Verbund zur Förderung des Hoch-und Höchstleistungsrechnens HLRN) and the Zentraleinrichtung für Datenverarbeitung (ZEDAT) at the Freie Universität Berlin for computational resources, the German Science Foundation (DFG) for funding within the CRC 1349—Fluorine-Specific Interactions.

**Conflicts of Interest:** The authors declare no conflict of interest.

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
