# Peer review of "Stability of Hydroxo/Oxo/Fluoro Zirconates vs. Hafniates—A DFT Study"

_inorganics, doi:10.3390/inorganics10120259_

Round 1

Reviewer 1 Report

The relative stability of Zr and Hf in their binary oxides, fluorides and oxofluorides have been systematically investigated by using DFT simulations. The structural parameters, reaction energies and some electronic properties are also explored to detect the affinity of Zr/Hf to O and F. The calculations show that the affinity of Zr/Hf to F is stronger than that to O. The manuscript is well organized and well written. However, I would not recommend this article for publication before the following comments are addressed.

(1)    It is well known that PBE functional is not appropriate for describing the interaction of d electrons. Therefore, in the current study the authors are suggest to convince the readers why they did not perform GGA+U calculations since Zr and Hf contain d electrons.

In the manuscript, the authors mentioned that the reaction energies obtained by the total energy calculated at 0 K is almost same as that free energy calculations by citing the Ref. 18. However, in the Table 1 of Ref. 18, one can clearly see that the reaction energy at 0K is higher than reaction free energy by energy values in a wide range, roughly 0.25~2eV. While in the current study, the reaction energy difference between Zr and Hf is just round 0.2~0.3 eV as shown in Table 2. Then the question automatically arises that relative trend of the affinity of Zr and Hf to O/F will probably be different if the temperature effect is considered. The authors are suggested to make comments or additional calculations.  

Author Response

Dear referee,
thank you for taking the time to evaluate our manuscript and for your suggested improvements. Please find our answer attached as pdf.

Reviewer 2 Report

The manuscript by Anders et al. reports the results of a theoretical investigation of  Zr/Hf binary oxides and fluorides, mono-hydroxylated fluorides,  as well as the hypothetical 1:2 mixed oxofluorides. The results shed some light on the subtle differences in the properties of the geochemical twin elements Zr/Hf, such as solubility products of the oxides in diluted aq. HF.

The manuscript may be of interest to researchers in the area of late transition metal chemistry and I have only several mostly technical remarks:

1. In the introduction part, please give a brief overview of what theoretical methods were already used for the same objects, what conclusions were drawn.

2. Please rationalize the choice of methods (functionals, plane waves) used for the calculations. Were different methods tested beforehand or the choice was based on literature?

3. The supplementary materials were missing in the submission, but are referenced in the manuscript.

4. Line 379 probably should read “solubility products of the oxides”

Author Response

Dear referee,

thank you very much for taking your time to review our manuscript. Please find our answer attached as pdf.

Reviewer 3 Report

Anders et al. presented their first-principles study on the stability of the Zirconates and Hafniates, and their Hydroxo/Oxo/Fluoro compounds. The topic relates to the chemistry behind the chemical subtractions for pure binary Hf- or/and Zr-compounds in practice. The code VASP and related GGA approximations are well-established. The settings seem OK. The design of the related chemical relations and crystal structures of the compounds is reasonable. The obtained information, including energetical preference of the Hf compounds to the corresponding Zr compounds might be of use for people working in the related fields. The manuscript is written properly. The text is in the scope of this Journal. Thus, I’d like to suggest acceptance of this manuscript for publication in Inorganics after some minor improvements.

1). It seems that this manuscript was prepared in a hurry. There are some typos and grammatical mistakes. I’d like to propose the authors spend more time and patience to improve the language in the text.

2). There are cases that the units of the data are not well-defined. For example, at Lines 141 and 143, the unit of the convergence energies should be specified to be per atoms or per unit cell.

Author Response

Dear referee,

thank you very much for taking your time to review our manuscript. We apologize for the typos and language errors and revised the full paper. All changes are highlighted in red.
We also thank you for pointing out, that the given convergence in energy per unit cell within the computational details was not clearly stated. Please find the according additions within revised manuscript. Accordingly, we also clarified per formula unit or per unit cell to all other given energies within the main manuscript and SI. 

Reviewer 4 Report

The authors of this paper worked on binary and ternary oxo/fluoro 1 compounds of the geochemical twin pair zirconium and hafnium. They tried to give some insights to the quantum-chemically-based hypothesis on the observed different solubility by solid Zr/HfO2 in aq. HF. However, though the work could be of interest to some researchers, I cannot recommend this paper for publication in its present form. Here is my comments:

1 .  The abstract is poorly written.  It should contain the importance or novelty of the work concisely.
2. Though the Introduction section is too big, still it is missing the importance of this particular work. The difference between this work and other published work is not mentioned in detail.

3.  What are the types of hydrogen bonding? What is the bond strength?

4 . What is the process of calculating the ΔEPos. of OH group?

5. How did the Bader charge was calculated and analyzed?

6. Is there any noncovalent interaction observed in the interface?

7. What is the underlying charge transfer mechanism for the reactions? 

Author Response

Dear referee,

thank you very much for taking your time to review our manuscript. We are thankful for your comments, which we considered in the following:
1. We have rephrased the abstract. All changes to the manuscript are highlighted in red.
2. We have adapted the introduction and included more information on other published work.
3. We are not certain what is meant by "types" of hydrogen bonding and "bond strength" other than the given information. If "bond strength" refers to an energy decomposition analysis, we would like to stress that assigning fragments inside a bulk material is highly ambiguous. Already in answer to 6. we did not calculate any interface or any adsorption. All calculations, also the mono-hydroxylations refer to bulk crystals.
4. To make ΔEPos. clearer, we adapted the table captions. It simply refers to the difference in total energy per unit cell among the different positional isomers.
5. As stated in the computational details line 160 (190 in revised manuscript), the “atoms-in-molecules-derived Bader charges have been obtained by the algorithm of the Henkelman group”. This is common practice for VASP calculations and the VASP website itself forwards to of the Henkelman group of the University of Texas  (https://vasp.at/info/resource/). Their Bader algorithm reads in the SCF-converged valence electronic density from the CHGCAR-file, that was calculated using the PAW potential files, as well as the reconstructed core and valence electronic densities from the files AECCAR0 and AECCAR2, which are generated by VASP when setting LAECHG = .TRUE..  Onto the reconstructed topology of the electronic density also close to the core regions, the same algorithm performs the actual atoms in molecules analysis searching for the planes of zero flux density and integrating the electronic density for each atomic Bader volume.
7. According to the calculated Bader charges, the charges within the binary or ternary oxo/fluorides hardly change. The tabulated reactions for cohesive energies from the elements should rather be considered as formal reactions. It is not our aim to model the mechanism of metallic Zr or Hf reacting with HF, H2O, O2 or F2. We rather aim to model the differences between Zr(IV) or Hf(IV) oxides/fluorides.

Round 2

Reviewer 1 Report

The authors have addressed all my concerns and I would like to recommend the article for publication without further changes.

Reviewer 4 Report

Authors of this work have considered some of my comments and revised their paper. Their argument on the calculation of Bader changes is not unacceptable. There are other population schemes that can be adopted without Mulliken charges, which I have already suggested.

Moreover, authors wrote that it is not their aim to model the mechanism of metallic Zr or Hf reacting with HF, H2O, O2 or F2. They rather aimed to model the differences between Zr(IV) or Hf(IV) oxides/fluorides. The latter part of their attempt does not include any physics of the systems explored; it is rather routine type, with no novelty. 

Furthermore, authors did not address my comments on a point-point-basis. The way they answered my questions is very unusual.